# Curcumin as an Epigenetic Therapeutic Agent in Myelodysplastic Syndromes (MDS)

**DOI:** 10.3390/ijms23010411

**Published:** 2021-12-30

**Authors:** Xiaoqing Xie, Daria Frank, Pradeep Kumar Patnana, Judith Schütte, Yahya Al-Matary, Longlong Liu, Lanying Wei, Martin Dugas, Julian Varghese, Subbaiah Chary Nimmagadda, Cyrus Khandanpour

**Affiliations:** 1Department of Medicine A, Hematology, Oncology and Pneumology, University Hospital Münster, 48149 Münster, Germany; Xiaoqing.Xie@ukmuenster.de (X.X.); Daria.Frank@ukmuenster.de (D.F.); pradeepkumar.patnana@ukmuenster.de (P.K.P.); Judith.Schuette@ukmuenster.de (J.S.); Longlong.Liu@ukmuenster.de (L.L.); lanying_imi@163.com (L.W.); chary.nimmagadda@ukmuenster.de (S.C.N.); 2Department of Hematology and Stem Cell Transplantation, University Hospital Essen, 45147 Essen, Germany; 3Department of Dermatology, University Hospital Essen, 45147 Essen, Germany; yahya.almatary@uk-essen.de; 4Institute of Medical Informatics, University Münster, 48149 Münster, Germany; Julian.Varghese@ukmuenster.de; 5Institute of Medical Informatics, Heidelberg University Hospital, 69117 Heidelberg, Germany; teamassistenz-postfach.imi@med.uni-heidelberg.de; 6Department of Hematology and Oncology, University Hospital of Schleswig-Holstein, University of Lübeck, 23562 Lübeck, Germany

**Keywords:** GFI1, acute myeloid leukaemia, histone acetyltransferase inhibitor, curcumin

## Abstract

Growth Factor Independence 1 (GFI1) is a transcription factor with an important role in the regulation of development of myeloid and lymphoid cell lineages and was implicated in the development of myelodysplastic syndrome (MDS) and acute myeloid leukaemia (AML). Reduced expression of *GFI1* or presence of the *GFI1-36N* (serine replaced with asparagine) variant leads to epigenetic changes in human and murine AML blasts and accelerated the development of leukaemia in a murine model of human MDS and AML. We and other groups previously showed that the *GFI1-36N* allele or reduced expression of GFI1 in human AML blasts is associated with an inferior prognosis. Using *GFI1-36S*, *-36N -KD*, *NUP98-HOXD13-tg* mice and curcumin (a natural histone acetyltransferase inhibitor (HATi)), we now demonstrate that expansion of *GFI1-36N* or *–KD, NUP98-HODXD13* leukaemic cells can be delayed. Curcumin treatment significantly reduced AML progression in *GFI1-36N* or *-KD* mice and prolonged AML-free survival. Of note, curcumin treatment had no effect in *GFI1-36S*, *NUP98-HODXD13* expressing mice. On a molecular level, curcumin treatment negatively affected open chromatin structure in the *GFI1-36N* or *-KD* haematopoietic cells but not *GFI1-36S* cells. Taken together, our study thus identified a therapeutic role for curcumin treatment in the treatment of AML patients (homo or heterozygous for *GFI1-36N* or reduced *GFI1* expression) and possibly improved therapy outcome.

## 1. Introduction

Acute myeloid leukaemia (AML) is the most common myeloid leukaemia in adults characterized by an accumulation of immature myeloid cells [1,2]. Myelodysplastic Syndrome (MDS), another disease of myeloid origin is characterized by the perturbed function of the myeloid-erythroid-megakaryocytic lineage resulting in neutropenia, thrombocytopenia and anaemia [3]. Approximately 30% of the MDS patients develop overt AML during the course of the disease [4]. Compared with other haematological malignancies, the prognosis of MDS/AML is relatively poor, since it is mainly a disease of elderly patients who cannot tolerate intensive chemotherapy regimens [2].

Growth Factor Independence 1 (GFI1) is a zinc finger transcriptional repressor and regulates the key function of hematopoietic stem cells (HSCs) [5,6] by recruiting histone deacetylases and histone demethylase complex to its target genes [7,8]. Murine studies have identified an important role for *Gfi1* in the quiescence of HSCs and their differentiation into cells of myeloid and lymphoid lineages [5]. Inherited mutation and variations in *GFI1* were reported in patients with severe congenital neutropenia [9]. Besides haematological malignancies, GFI1 mutation or reduced GFI1 expression enhances the progression of other cancers such as colorectal [10], lung [11], prostate, and breast cancers [12] Besides haematological malignancies, GFI1 mutation or reduced GFI1 expression enhances the progression of other cancers such as colorectal [10], lung [11], prostate, and breast cancer [12].We have formerly reported a single nucleotide polymorphism (SNP) in *GFI1* (*GFI1-36N*, 36 serine (S) replaced with asparagine (N)) and its role in the induction of *Hoxa9* locus and acceleration of K-RAS driven myeloproliferative disorder in mice [13]. Subsequently, we generated various murine strains characterized by lower GFI1 expression or those expressing cDNA of full-length human *GFI1-36S* or *-36N* in murine locus and demonstrated their role in MDS and AML [6,13,14]. The mouse models elucidated an important role for reduced Gfi1 expression and the expression of *GFI1-36N* SNP variant in the accelerated onset of fatal myeloid proliferative disease (MPS) and increased the risk for AML progression [6,7,14,15].

Epigenetic modifications lead to alteration in the expression of genes and have been implicated in AML pathogenies [16]. Epigenetic modifying agents such as DNA hypomethylation agents (like azacytidine and decitabine) are widely used to treat MDS and AML patients [17] and there are several ongoing preclinical and clinical studies testing histone deacetylase (HDACs) inhibitors (like panobinostat and vorinostat) in myeloid malignancies [18]. *GFI1-36N* or reduced expression of *GFI1* lead to epigenetic changes, potentially due to their reduced ability to recruit histone-modifying enzymes (HDACs 1-3) to their target genes [19]. Supporting these observations, data from our group also demonstrated that reduced expression of GFI1 protein or *GFI1-36N* allele in murine AML blasts led to an increase in H3K9 acetylation at GFI1 target genes [6,14]. Furthermore, histone acetyltransferase inhibitors (HATi), CTK7a treatment blocked leukaemia progression in healthy mice transplanted with *GFI1-36N* homo or heterozygous murine AML cells [6,14].

Curcumin a polyphenolic compound derived from *Curcuma longa* is characterized by potent anti-oxidant and anti-inflammatory properties by modulating cytokines like TNF-α or interleukin-6 [20,21]. Additional, curcumin has been shown to have membrane protective, antimicrobial, thrombosis supressing as well as antirheumatic effects [22,23].Curcumin is a natural HATi with no associated toxicities even at high doses (12 g/day) [22,24] and is often used as food additive especially in Asian countries [20]. In the past twenty years curcumin has gotten a lot of attention due to its anti-tumour effects. Recent studies have found its effect on several signalling pathways such as cell cycle, NF-κB signalling and mitogen-activated protein kinase (MAPK) pathway leading to efficient induction of proliferation and provoked apoptosis [21,23]. Studies have confirmed its anticancer activities in various cancers like prostate and colorectal cancer as well as AML [20,21,25].Current therapy with DNA demethylating agent Azacitidine and chemotherapeutic drugs are characterized by a wide side effect profile and limited curative therapeutic regimens. Employing our mouse models differentially expressing GFI1, we were thus interested in evaluating the therapeutic potential of curcumin in a setting of MDS-AML patients expressing reduced GFI1 or *GFI1-36N* SNP variant.

## 2. Results

### 2.1. Mice AML-Free Survival in Curcumin Un-Treatment and Curcumin Treatment

To investigate our hypothesis that curcumin treatment could therapeutically benefit MDS patients with lower expression of GFI1 or those expressing *GFI-36N*, we used *NUP98-HOXD13* transgenic (*tg*) mice as a model for MDS. The fusion gene *t*(*2*;*11*) (*q31*;*p15*) characteristic of the mouse model lead to the development of highly penetrant disease similar to human MDS with blast cells in the bone marrow (BM), dysplastic features of myeloid cells, and progressing to cytopenia [14]. A fraction of these mice developed leukaemia reminiscent of AML with characteristics consistent with human MDS as reduced BM infiltration, the appearance of blast cells in the peripheral blood (PB) and succumbed to leukaemia from cytopenia. Using this model we have demonstrated that the reduced expression of GFI1 or presence of *GFI1-36N* accelerated progression to AML in the *NUP98-HOXD13* murine model with MDS [6,26]. The presence of *GFI1-36N* or reduced expression of *GFI1* (*GFI1-KD*) shortened the latency period and increased the incidence of AML [1].

Following the previously described strategy [1,6,14], *GFI1-36S*, *-36N*, and *-KD*, *NUP98-HOXD13-tg* mice were treated with vehicle or curcumin. All the mice were characterized with no obvious differences in overall survival but a difference in latency and frequency of AML development (Figure 1A–D and Appendix Appendix A). Interestingly, curcumin treatment of *GFI1-36N* and -*KD*, *NUP98-HOXD13-tg* mice impeded AML progression significantly leading to a prolonged AML-free survival time (Figure 1B–D). Overall, curcumin treatment impeded AML development in *GFI1-36N* and -*KD*, *NUP98-HOXD13-tg* mice but had no effect in *GFI1-36S*, *NUP98-HOXD13-tg* mice (Figure 1D). Different studies have shown that c-kit positive (ckit+) leukemic cells are enriched for leukemic stem cells [27]. We, therefore, quantified ckit+ cells to gain further insights into leukaemogenesis and survival in curcumin-treated mice. Curcumin treatment of *GFI1-36N* and *-KD*, *NUP98-HOXD13-tg* mice led to a decrease in the percentage of c-kit+ cells in the BM but not in the control *GFI1-36S*, *NUP98-HOXD13-tg* mice (Figure 2A, 77.8% to 24.2% in *GFI1-KD* setting, *p* < 0.001 and 78.3% to 31.8% in *GFI1-36N* mice, *p* < 0.001). However, no changes in the percentage of c-kit+ cells in *GFI1-36S*, *NUP98-HOXD13-tg* mice were observed in BM (Figure 2A). Similar observations were made in the spleen. Percentage of the c-kit+ cells were decreased in *GFI1-KD* and *-36N, NUP98-HOXD13-tg* mice (Figure 2B, 60% to 23.7%, *p* < 0.005 and from 64.3% to 32.1%, *p* < 0.01 respectively). Again, no changes in the percentage of *GFI1-36S*, *NUP98-HOXD13-tg* mice were observed. We next investigated additional parameters as platelet (PLT) number and haemoglobin (Hb) in various settings. In *GFI1-36S*, *-36N* or *GFI1-KD*, *NUP98-HOXD13-tg* mice, curcumin treatment did not alter Hb or PLT number (Figure 2C,D). BM cytospin analysis from different mice indeed confirmed a role for curcumin treatment in the suppression of leukemic blasts in *GFI1-36N* and *-KD*, *NUP98-HOXD13-tg* leukemic mice (Appendix Appendix AA,B). This is in line with our earlier observations underscoring a role for curcumin treatment in improved AML free survival in mice (Figure 1A–D). In summary, our observations indicated that curcumin treatment in line with our hypothesis impeded the progression of MDS to AML and was associated with a lower tumour burden in *GFI1-36N*, *-KD, Nup98-HOXD13-tg* mice.

### 2.2. Expression of GFI1-36N and GFI1-KD Induced Differential Epigenetic Changes as Compared to GFI1-36S

We have earlier reported that expression of *GFI1-36N* or a lower expression of *GFI1* in leukemic cells leads to higher levels of H3K9 acetylation around transcription start sites (TSSs) of previously identified GFI1 target genes in comparison to *GFI1-36S* leukemic cells [1,14]. We, therefore, sought to analyse the effect of curcumin on these genes by performing ChIP-seq analysis with a focus on the H3K9 acetylation on GFI1 target genes. Our analysis indeed revealed that the curcumin treatment of *GFI1-36N* or *–KD*, *Nup98-HOXD13* lead to lower acetylation of the global promotor region (+/− of 1 kb from TSS), as compared to leukemic cells expressing *GFI1-36S* (Figure 3B,C). As evident by these data, the average distribution of H3K9ac relative to the TSS across genes in leukemic cells from *GFI1**-36N* and *-KD* was lowered and this was even further pronounced in *GFI1-KD* blast cells with a lower log2 fold change (Figure 3B,C). Curcumin treatment however had a minimal effect on acetylation in *GFI1-36S* leukemic cells (Figure 3A). Furthermore, GSEA analysis of the data identified a negative enrichment for tumour related pathways in *GFI1-36N* and -KD mice blast cells treated with curcumin (Figure A4), potentially explaining the factors contributing to the lower AML incidence (Figure 1D), lower c-kit+ cells in BM and spleen (Figure 2A,B), and the number of blast cells (Appendix Appendix AA,B). At a conventional threshold of *p* < 0.05 and an FDR value of <0.25, hallmark_heme metabolism, in particular, has drawn our attention since it was negatively enriched in both *GFI1-36N* and *-KD* curcumin-treated leukemic mice but positively enriched in curcumin-treated *GFI1-36S* mice (Figure 4). Our findings thus identified heme metabolism as one of the possible factors governed by GFI1 expression and need further investigation. In summary, our data thus underscored a role for different forms of GFI1 protein in histone acetylation and thereby regulating GFI1 expression dependent TSS and AML development.

## 3. Discussion

We have earlier reported that a lower expression of GFI1 or the expression of its SNP variant, promoted the progression of MDS to AML in the murine model of MDS/AML, *NUP98-HOXD13* [1,6,14]. At a molecular level, as compared with the *GFI1-36S* murine model, *GFI1-36N* or lower expression resulted in increased H3K9 acetylation of the regulatory elements of GFI1 target genes and contributed to the progression of MDS to AML [7]. Complementing this observation, histone acetyltransferase inhibitor (HATi) treatment with CTK7a impeded leukaemia progression of *GFI1-36N* expressing leukemic cells [1,6,14]. CTK7a is a nonspecific, toxic inhibitor and is unavailable for use in clinics. We, therefore, sought to investigate curcumin (a natural antioxidant, anti-inflammatory and a HATi) for its efficacy in impeding the MDS-AML progression, improving therapy outcome and shedding light on the molecular basis of the effect of curcumin treatment.

Using the *NUP98-HOXD13* murine model of MDS, our data demonstrated that curcumin treatment specifically impeded MDS-AML progression. As indicated by different markers (frequency of c-kit+ cells in spleen and BM, reduced blast counts) diseases progression was hindered in *GFI1-36N* and *-KD, NUP98-HOXD13-tg* mice treated with curcumin. On a molecular level, CHIP sequencing in curcumin-treated *GFI1-36N* and *-KD, NUP98-HOXD13-tg* mice, demonstrated an upregulation of H3K9 acetylation (open chromatin sites) but not in control mice treated with vehicle or curcumin. Of note, these sites were of relevance and have been implicated in AML and cancer development. To gain further insights into different signalling cascades, a GSEA analysis was performed. Interestingly, the hallmark_heme metabolism was negatively enriched in both *GFI1-36N* and *-KD, NUP98-HOXD13-tg* mice but positively enriched in curcumin-treated *GFI1-36S* mice. This observation potentially indicated a role for different forms of GFI1 protein in regulating reduced histone acetylation and thereby affecting diversified processes driving AML progression. The negative effects of curcumin treatment in the reduction of intracellular heme level have been reported in SH-SY5Y (human neuroblastoma) cells by upregulating heme oxygenase 1 (HO-1), a rate-limiting enzyme responsible for heme degradation. [28]. Recently in paediatric AML patients, upregulated *MYCN* could be associated with increased heme biosynthesis, which is required for self-renewal and maximal mitochondrial respiration [29]. Furthermore, the study also identified therapeutic vulnerabilities by targeting porphyrin homeostasis, preventing *MYCN* hematopoietic progenitor self-renewal and suppressing myeloid leukaemogenesis [29]. Interestingly the paralogue of *Gfi1, Gfi1aa* and *Gfi1b*, have been implicated in setting the pace for primitive red blood cells differentiation from hemangioblasts in zebrafish embryos [30]. Furthermore, in the context of B cell lineage leukaemia’s, the heme-sensing pathway modulated apoptotic sensitivity by repressing *MCL-1* and increased their responsiveness to BH3-mimetics [31].

However, the experiments in this study were performed with only one murine model of MDS as others were not available to us. Preclinical studies with human MDS in murine setting are difficult to establish but this would represent future directories. Due to technical limitations, we could not correlate epigenetic changes with gene expression changes.

Taken together, our observations improve our understanding of GFI1 function, identified a novel role for GFI1 variants in heme biosynthesis and established GFI1 variants as biomarkers in AML therapeutic strategies. However further studies are warranted to better understand the role of different GFI1 proteins in regulating heme metabolism and to exploit at a therapeutic level in AML patients. This study shows that additional epigenetic therapeutic approaches could improve prognosis of MDS patients and since curcumin is well tolerated this would justify a Phase I trial with MDS patients.

## 4. Materials and Methods

### 4.1. Mouse Strains

*NUP98-HOXD13* transgenic (*tg*) mice were obtained from The Jackson Laboratory (Bar Harbor, ME, USA). *GFI1-36S*, *-36N* and *-KD* mouse trains have been previously described [1,6,14]. The mice were housed under specific pathogen-free conditions. Animal studies were performed following the approved protocols of the government ethics committee for animal experimentation (document numbers 84-02.04.2015.A022 and 84-02.04.2015.A076).

### 4.2. Curcumin Treatment

Curcumin (curcuma longa) was procured from Sigma (Cat. C1386-50G Missouri, United States). The mice were treated with 20 mg/kg curcumin weekly by intraperitoneal (i.p.) injection. Control mice received an equal volume of DMSO.

### 4.3. Flow Cytometry

When the mice showed signs of overt leukaemia or reduced overall health, mice were euthanized and cells from BM, spleen, PB cells were analyzed. Various antibodies to evaluate onset of leukaemia were procured from BioLegend (San Diego, CA, USA) (APC anti-mouse CD117 (c-kit), PE anti-mouse Ly-6G/Ly-6C (Gr-1), PerCP anti-mouse (CD11b), PE anti-mouse (CD8a), PerCP anti-mouse (CD4), PerCP anti-mouse (B220), PE anti-mouse (TER-119). Apoptosis was quantitatively analysed by flow cytometry using Annexin V (BioLegend, San Diego, CA, USA) and propidium iodide (Sigma, MO, USA). The flow cytometry staining was performed as previously described [6].

### 4.4. Chromatin Immunoprecipitation Sequencing (ChIP-Seq)

ChIP assays were performed as previously described [32]. Briefly, 1 × 10^7^ frozen BM cells were washed with PBS. The cells were cross-linked with 1% formaldehyde, lysed and sonicated to an average size of around 500 bp. Immunoprecipitation was conducted anti-histone H3 (acetyl 9) antibody (Abcam ab4441) or a control antibody and collected with protein G beads. The beads were washed and samples were eluted. After reverse-crosslinking, eluted DNA was purified with the Qiaquick PCR Purification Kit (QIAGEN, Germantown, Maryland). Following the manufacturing instructions, we amplified each sample using the Illumina kit and sequenced it with the Illumina 2G Genome Analyzer. Furthermore, Trim Galore was used to remove adapters from the sequencing reads. Sequencing reads were mapped to the mouse reference genome mm10 using BWA-MEM. Peaks were called using MACS2 and filtered with ENCODE black list regions. Peaks were then annotated by ChIPseeker. DiffBind was used for consensus peak finding and read count normalization. Enhancers were annotated based on curated enhancer regions from EnhancerAtlas 2.0.

### 4.5. Gene Expression

Gene set enrichment analysis (GSEA) was performed by comparing *GFI1-36S/ -36N/ -KD* samples isolated from with curcumin and vehicle-treated mice and differentially regulated Hallmark gene sets in the Molecular Signature Database were analyzed. The input of GSEA is the normalized peak read counts of all genes. For genes with multiple associated peaks, the peak with the largest absolute fold change between two conditions was used. GSEA was set to rank genes based on the log2 fold change of the normalized peak read counts of two conditions.

### 4.6. Statistical Methods

Statistical analyses were performed using GraphPad Prism 9 software (GraphPad Software, La Jolla, CA, USA). Paired or unpaired two-sided *t*-test, Log-rank (Mantel-Cox) test were used to analyse statistical significance. A *p*-value of <0.05 was considered significant.

## Figures and Tables

**Figure 1 ijms-23-00411-f001:**
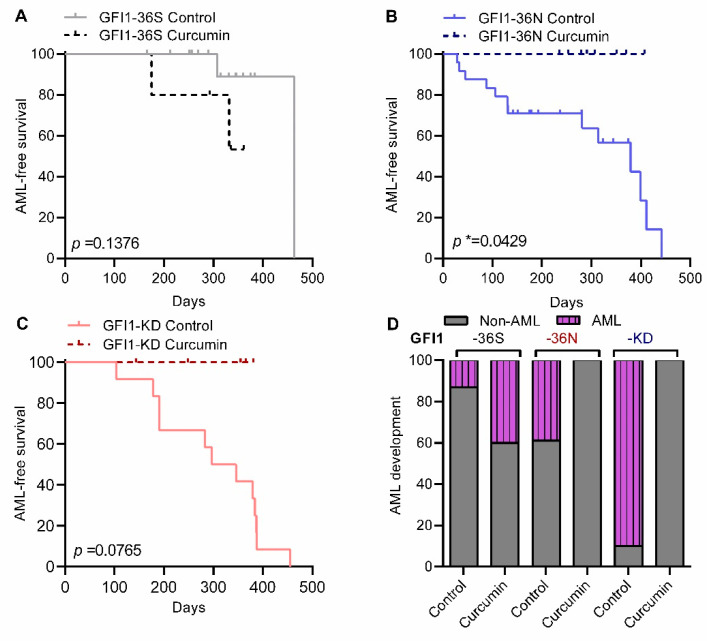
Effect of curcumin on AML free survival of *GFI1-36S/ -36N/-KD*, *NUP98/HOXD13*-*tg* mice. (**A**–**C**). Overview of the AML-free survival in GFI1-36S (**A**), GFI1-36N (**B**) and GFI1-KD (**C**) mice treated with curcumin or vehicle. (**D**). Quantification of AML development in vehicle or curcumin-treated *GFI1-36S*, *-36N* and *-KD, GFI1-36S/ -36N/-KD*, *NUP98/HOXD13*-*tg* mice. * *p* < 0.05.

**Figure 2 ijms-23-00411-f002:**
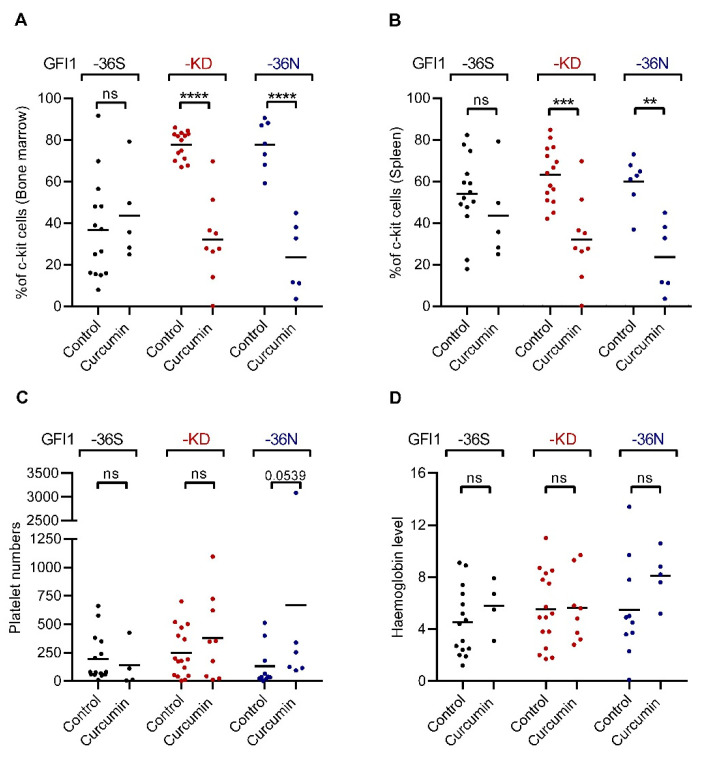
Curcumin treatment affected parameters contributing to AML development. (**A**,**B**). Percentage of c-kit positive (c-kit+) cells in the bone marrow (BM) and spleen in the vehicle and curcumin-treated *GFI1-36S/ -36N* and *-KD, NUP98/HOXD13-tg* mice. (**C**,**D**). Quantification of platelets and haemoglobin in curcumin and vehicle-treated *GFI1-36S/ -36N* and *-KD, NUP98/HOXD13-tg* mice. **** *p* < 0.001, *** *p* < 0.005, ** *p* < 0.01. ns: not significant.

**Figure 3 ijms-23-00411-f003:**
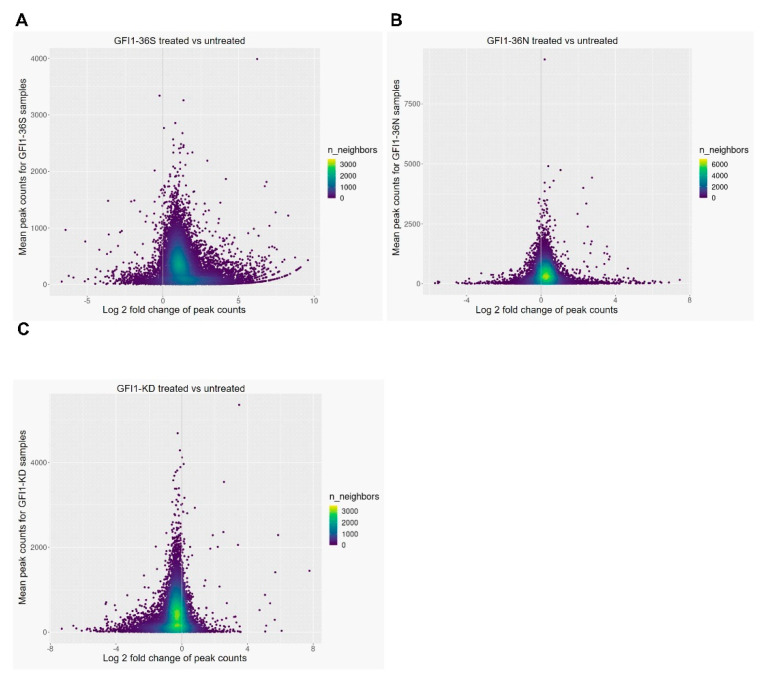
Curcumin treatment affected GFI1 promoter occupancy in the c-kit+ cells *GFI1-36S/-36N* and *-KD, NUP98/HOXD13-tg* mice. CHIP sequencing was performed in the frozen BM cells of *GFI1-36S/ -36N* and *-KD, NUP98/HOXD13-tg* mice. The mean and log2 fold change of peak counts (promoter ±1 kb) in *GFI1-36S* (**A**), *-36N* (**B**) and *–KD* (**C**), *NUP98/HOXD13-tg* mice (vehicle vs curcumin treatment) are shown. Peak counts were calculated as normalized read counts for each consensus peak located in promoter regions (±1 kb). Individual points are coloured by the number of neighbouring points to display the point density.

**Figure 4 ijms-23-00411-f004:**
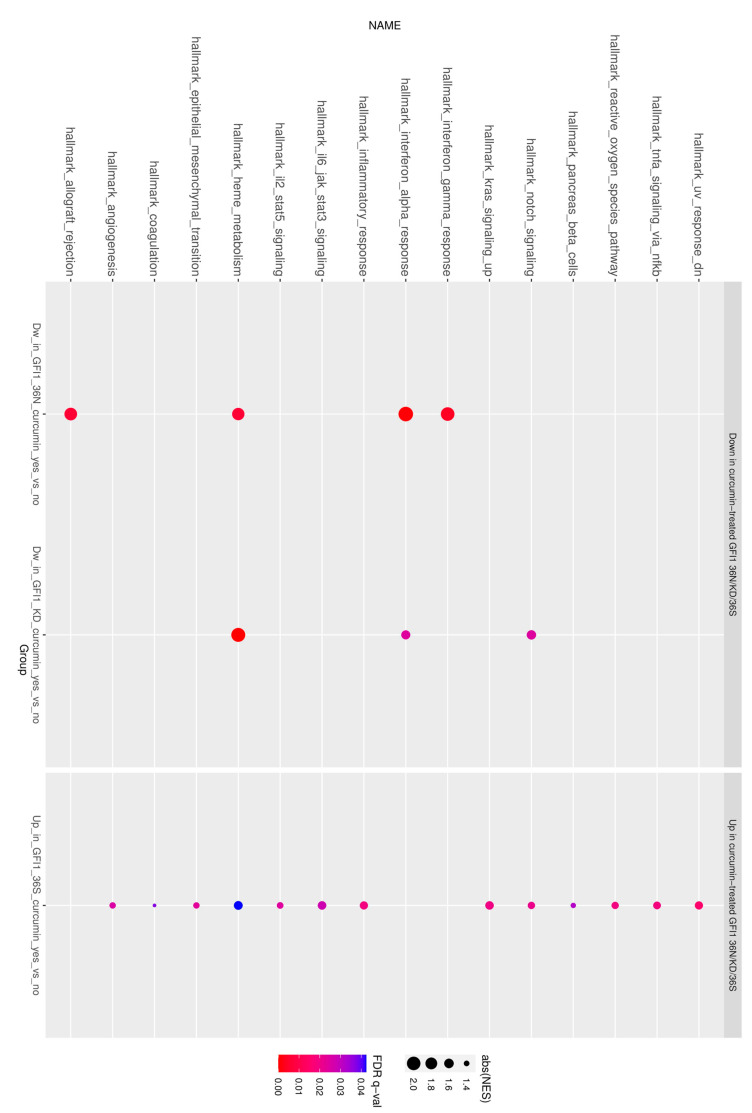
Curcumin treatment significantly affected hallmark Heme-metabolism in *GFI1-36N/-KD, NUP98-HOXD13-tg* mice. Significantly enriched (FDR q-value < 0.05) hallmark gene sets between curcumin-treated and untreated samples are presented. Genes were ranked by log 2 fold change of normalized peak read counts between curcumin-treated and untreated samples. For genes with multiple associated peaks, the peak with the largest absolute fold change was used. The colour of the dot denotes the FDR q-value and the size of the absolute value of the normalized enrichment score. The columns from left to right respectively demonstrate negatively enriched gene sets in GFI1-36N/ -KD and positively enriched gene sets in *GFI1-36S*, *NUP98-HOXD13-tg* curcumin-treated samples.

## Data Availability

ChIP-seq data from murine bone marrow samples are deposited under GEO GSE 190974. https://www.ncbi.nlm.nih.gov/geo/info/linking.html. (now this is a private link, it’s accessed date: 1 January 2022).

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
