# Peer review of "Curcumin as an Epigenetic Therapeutic Agent in Myelodysplastic Syndromes (MDS)"

_ijms, 2021, doi:10.3390/ijms23010411_

Round 1

Reviewer 1 Report

The manuscript was prepared very well. The introduction section justifies the purpose of the study. I congratulate the authors for the preparation of the manuscript

I would like to congratulate the authors for the structure of the manuscript and all the research carried out. It is highly publishable. However, there are some concerns, in part important, so the review articles need revision, see below.

Introduction

  • It should include a more extensive description of curcumin, including properties. You can use doi:10.3390/nu12020501
  • Which is the figure that appears at the beginning of the introduction, lacks a title and is not reflected in the text.

Materials and Methods

The methodology is perfectly described and carried out

Results

  • The tables and the text describing them do not require any input, it is the strongest part of this study.

Discussion

  • Include a limitations section.
  • what does this manuscript specifically contribute?

Author Response

Introduction

Comment 1: It should include a more extensive description of curcumin, including properties. You can use doi:10.3390/nu12020501

Answer 1: We included some additional and more detailed information about curcumin in the introduction. We used amongst others the suggested reference. See page 3 line 82-92.

Comment 2: Which is the figure that appears at the beginning of the introduction, lacks a title and is not reflected in the text.

Answer 2: This figure is showing a graphical abstract of the study. We now included a headline “graphical abstract”. See page 2 line 39.

Discussion

Comment 3: Include a limitations section.

Answer 3: We have performed this study with only one murine model of MDS as others were not available to us. Preclinical studies with human MDS in murine setting are difficult to establish but this would represent future directories. Due to technical limitations, we could not correlate epigenetic changes with gene expression changes. However, the biological readout is quite strong and supports further studies. A limitation section was added at page 8 line 236-239

Comment 4: What does this manuscript specifically contribute?

Answer 4: It shows that additional epigenetic therapeutic approaches could improve prognosis of MDS patients and since curcumin is well tolerated this would justify a Phase I trial with MDS patients. This part was added at page 8 line 244-246.

Reviewer 2 Report

The manuscript entitled “Curcumin as an epigenetic therapeutic agent in Myelodysplastic syndromes (MDS)” by Dr. Xie and colleagues evaluated the antineoplastic potential of curcumin, an acetyltransferase inhibitor, in an acute myeloid leukaemia (AML) mouse model comprising GFI1-36S, -36N -KD, NUP98-HOXD13-tg mice. Main results indicate that curcumin treatment can significantly reduce AML progression in GFI1-36N or -KD mice and prolonged AML-free survival, while having no effect in GFI1-36S, NUP98-HODXD13 expressing mice. Moreover, curcumin treatment affecteded open chromatin structure in the GFI1-36N or -KD haematopoietic cells but not GFI1-36S cells. This is an interesting work reporting important data obtained in vivo with mice models on the role of Curcumin as therapeutic agent AML. The reported data are clearly present and well discussed. The ms is in general well written and sections well organized, while it might reach interest across the readers. However, figures should be improved.

While recommending a minor revision, I have a several observations for improving the ms

General comments
•    There is a figure without label and caption and probably in the wrong position at page 2. Please fix the problem
•    The quality of the images in general should be improved

Minor observations
Lines 42-56 a role for GFI1 in cancer as tumor suppressor gene has also been reported in other tumor models, such as Endocrine-Related Cancers PMID: 32630147 and Colorectal Cancer PMID: 30606770. This information should be included
Line 58 Consistently, epigenetic agents, such as DNA hypomethylating agents and/or histone deacetylase inhibitors, have broadly been employed for antitumor therapies in Myelodysplastic syndromes (PMID: 33958699 and PMID: 30304859). This information should be included as a background.
Line 118 pelase include p values in panels A B and C
Line 123 statistical comparisons should be included in all panels from figure 2
Lines 198-200 The cell type should be detailed
Lines 227-233 please include references supporting the mice cell isolation/Flow cytometry methods

Author Response

General comments:

Comment 1: There is a figure without label and caption and probably in the wrong position at page 2. Please fix the problem

Answer 1: This figure is showing a graphical abstract of the study and therefore we kept it after the written abstract. We now included a headline “graphical abstract”.

Comment 2: The quality of the images in general should be improved

Answer 2: We improved the quality of all the images to a resolution of 1200 dpi.

Minor observations:

Comment 3: Lines 42-56 a role for GFI1 in cancer as tumor suppressor gene has also been reported in other tumor models, such as Endocrine-Related Cancers PMID: 32630147 and Colorectal Cancer PMID: 30606770. This information should be included

Answer 3: We added other cancers (e.g. breast, lung cancer) were reduced GFI1 or mutation of GFI1 promotes cancer progression. See page 2 line 55-57.

Comment 4: Line 58 Consistently, epigenetic agents, such as DNA hypomethylating agents and/or histone deacetylase inhibitors, have broadly been employed for antitumor therapies in Myelodysplastic syndromes (PMID: 33958699 and PMID: 30304859). This information should be included as a background.

Answer 4: We included some epigenetic agents used in the clinic to treat MDS patients or patients with myeloid malignancies. See page 3 line 70-74.

Comment 5: Line 118 please include p values in panels A B and C

Answer 5: We included the p values for panel A, B and C from figure 1. See page 4 figure 1.

Comment 6: Line 123 statistical comparisons should be included in all panels from figure 2

Answer 6: We included a statistical comparison for all panels and included ns for non-significant. See page 5 figure 2.

Comment 7: Lines 198-200 The cell type should be detailed

Answer 7: In the described study, they used SH-SY5Y (neuroblastoma cell line) cells. We added this information at page 8 line 221-224.

Comment 8: Lines 227-233 please include references supporting the mice cell isolation/Flow cytometry methods

Answer 8: We included one reference at page 9 line 267-268. Our group is using this flow cytometry staining method since long and several papers has been published in which we used the same antibodies to analyze the spleen, bone marrow and peripheral blood cells of transplanted mice.